# Snowfall distribution and its response to the Arctic Oscillation: An evaluation of HighResMIP models in the Arctic using CPR/CloudSat observations

Manu Anna Thomas[1], Abhay Devasthale[1], Tristan L'Ecuyer[2], Shiyu Wang[1], Torben Koenigk[1, 3], and Klaus Wyser[1]

[1]Swedish Meteorological and Hydrological Institute, Folkborgsvägen 17, 60176 Norrköping, Sweden.
[2]University of Wisconsin-Madison,1225 West Dayton Street, Madison, WI 53706.
[3]Bolin Centre for Climate Research, Stockholm University, 10691 Stockholm, Sweden.

**Correspondence:** Manu Anna Thomas (manu.thomas@smhi.se)

**Abstract.** A realistic representation of snowfall in the general circulation models (GCM) is important to accurately simulate snow cover, surface albedo, high latitude precipitation and thus the surface radiation budget. Hence, in this study, we evaluate snowfall in a range of climate models run at two different resolutions using the latest estimates of snowfall from CloudSat Cloud Profiling Radar over the northern latitudes. We also evaluate if the finer resolution versions of the GCMs simulate the accumulated snowfall better than their coarse resolution counterparts. As the Arctic Oscillation (AO) is the prominent mode of natural variability in the polar latitudes, the snowfall variability associated with the different phases of the AO is examined in both models and in our observational reference. We report that the statistical distributions of snowfall vary considerably between the models and CloudSat observations. While CloudSat shows an exponential distribution of snowfall, the models show a Gaussian distribution that is heavily positively skewed. As a result, the 10 and 50 percentiles, representing the light and median snowfall, are overestimated by up to a factor of 3 and 1.5 respectively in the models investigated here. The overestimations are strongest during the winter months compared to autumn and spring. The extreme snowfall represented by the 90 percentiles, on the other hand, is positively skewed underestimating the snowfall estimates by up to a factor of 2 in the models in winter compared to the CloudSat estimates. Though some regional improvements can be seen with increased spatial resolution within a particular model, it is not easy to identify a specific pattern that hold across all models. The characteristic snowfall variability associated with the positive phase of AO over Greenland Sea and central Eurasian Arctic is well captured by the models.

## 1 Introduction

Snowfall is one of the key geophysical variables in the Earth System. From the climate perspective, snowfall regulates the surface albedo and air-surface interactions, thus playing a key role in the radiation budget over the high latitude regions. Up to 80-90% of incoming shortwave solar radiation is reflected by snow covered surfaces during winter (Geiger, 1957; Barry, 1996). At the same time, snow cover acts as an excellent thermal insulator (Mellor, 1964; Sturm et al., 1997) in winter when the

radiation balance is dominated by longwave radiation losses to space. Loss of highly reflective Arctic snow/ice surfaces would open up more darker land or ocean surfaces and hence, enhance surface warming by increasing the absorption of sun's energy. Snowfall is also the dominant form of precipitation in the polar regions and an important component of the hydrological cycle in the high latitude regions during the winter half year. Most recently, Li et al. (2018) argued that the greenhouse effect of the

falling snow (longwave forcing) is an important process and can potentially help explain the underestimated rate of sea-ice decline in climate models.

From the weather perspective, snowfall is also a key variable, especially, since it has a socio-economic dimension. For example, snowfall events have an impact on air and surface traffic and winter tourism at the local level. The lack of sufficient snowfall at ski resorts can have large economic costs associated with it locally. The lake-effect snowfall is another example that

can have catastrophic impacts on resources planning and economic costs down to a district level, as the large amount of snow is deposited in a short time. Furthermore, heavy snowfall events are linked to health concerns, such as heart attacks, especially in the elderly and vulnerable population (Auger et al., 2017).

In light of these multiple effects of snowfall, a good quantitative understanding of snowfall amount and its intraseasonal and interannual variability is needed, not the least to address key scientific questions related to future changes in the Earth

System. For example, as the Arctic experiences the amplified surface warming and rapidly decreasing sea-ice, how would this new paradigm affect the snowfall (and thus the hydrological cycle) in the Arctic? How would the changes in the Arctic climate system affect heavy or extreme snowfall events in the mid- to high latitude regions? Recent studies indicate decreasing trend in snow cover over the Arctic, later snow cover onset and earlier snow-free dates and decrease in snow cover duration (Liston and Hiemstra, 2011; Callaghan et al., 2011). The declining Arctic sea-ice is linked to the heavy snowfall events over the large parts

of northern hemispheric continents during recent winters (Liu et al., 2012). Also, studies have shown that the Arctic Oscillation (AO), the main index of circulation in the Arctic, tends to be in a positive phase since the last two decades, thereby resulting in an increase in the winter precipitation in northern Eurasia and a decrease over southern Eurasia and north-east Canada (Givati and Rosenfeld, 2013; Qu et al., 2015; Gong et al., 2014). The changes in the Arctic climate system have further implications for the mid-latitude weather systems, including snowfall (Cohen et al., 2014, 2018).

In order to grasp a better understanding of such key processes and, more importantly, to be able to predict future changes in snowfall and associated feedbacks, both reliable observations of snowfall and high fidelity global climate models are needed. Direct observations of snowfall have been very difficult in the past. Most precipitation observations are available on land. Gridded snowfall observations over the polar and oceanic regions are lacking. As a result, the snowfall variability and trends are often studied based on the reanalysis datasets and models that determine snowfall from climatology of temperature and

precipitation (Roesch, 2006; Krasting et al., 2013). In contrast to the snowfall amount, the snow cover observations are available from a number of passive satellite sensors with better spatio-temporal coverage (Bokhorst et al., 2016).

With the launch of active Cloud Profiling Radar (CPR) onboard NASA's CloudSat satellite since 2006, realistic estimates of global snowfall amounts are possible. For example, Kulie et al. (2016) provided a first near-global survey of snowfall from shallow cumulus systems during CPR/CloudSat retrievals. Most recently, Palerme et al. (2017) evaluated how well the promi-

nent global reanalysis datasets represent snowfall over Antarctica. With a record spanning more than a decade, CPR/CloudSat provides unprecedented opportunity to statistically evaluate snowfall in global climate models.

The lack of snowfall observations in the past have meant that the snowfall processes in global models are not likely to be represented with high fidelity. Indeed, over the greater Alpine region, Terzzago et al. (2017) have shown considerable differences between snowfall observations and model simulations from the latest-generation regional and global climate models (RCMs, GCMs), participating in the Coordinated Regional Climate Downscaling Experiment over the European domain (EURO-CORDEX) and in the Fifth Coupled Model Intercomparison Project (CMIP5) (Jacob et al., 2014; Taylor et al., 2012).

The atmosphere-ocean coupled climate models from the CMIP5 indicate a snowfall redistribution in the Northern Hemisphere in future climate scenarios (Krasting et al., 2013). As the greenhouse gases and surface temperature increases in the Arctic are expected to continue atleast a few more decades, studying snowfall-climate interactions becomes even more important. However, the first step in this direction is to evaluate the fidelity of climate models in simulating spatio-temporal distribution of snowfall using observations. Such detailed evaluation of GCMs likely to participate in the next IPCC assessments, using the latest CloudSat observations, over the northern high latitudes, including the Arctic, is currently lacking. Hence, the main aim of this study is to evaluate the HighResMIP (High Resolution Model Intercomparison Project) simulations for CMIP6 (Haarsma et al., 2016) under the PRIMAVERA (PRocess-based climate sIMulation: AdVances in high resolution modelling and European climate Risk Assessment) project. This project is a European Union H2020 project wherein a total of 7 state of the art models are run at varying resolutions to understand the impact of resolution on different global climate processes. In this context, in the present study, we address the following questions. 1. How well do the GCMs used in the framework of the EU PRIMAVERA project simulate the northern high latitude snowfall? 2. Does increasing the spatial resolution improve the snowfall representation in these models? and 3. Do the models simulate the snowfall variability associated with the different phases of Arctic Oscillation (AO) realistically?

## 2  Models, observations and methodology

### 2.1  Models participated in the PRIMAVERA project

The snowfall from four HighResMIP models at different resolutions is evaluated against observations. The table below (Table. 1) gives a brief description of the models that were used in this study. All models that are evaluated here are atmosphere-only models that are forced with HadlSST2.2 (Kennedy et al., 2017) and sea ice concentrations and are run at two different horizontal resolutions. Since the focus is on northern high latitudes (beyond $50^o$N latitude), the models can be classified clearly into high (Hi-res) and low/coarse (Lo-res) set-ups with high/low resolution set-ups having a resolution below/above 35 kms. The model simulations cover the satellite period from 1980 until 2015. The background aerosol climatology varies from model to model. However, the anthropogenic aerosol forcing is generated by the MACv2-SP method (Stevens et al., 2017), wherein the aerosol forcing is calculated based on the aerosol optical properties and fractional change in cloud droplet number concentrations. External forcings follows the HighResMIP protocol described in Haarsma et al. (2016). The state-of-the-art climate models feature prognostic microphysics schemes with several species of condensed water, typically liquid and ice cloud water, rain,

snow and possibly, also graupel. All processes related to water clouds - CCN activation, autoconversion and accretion - are relatively well known and also well represented in climate models. Ice processes, on the other hand, are far less explored and the parameterisatons for ice nuclei activation or aggregation of ice cristals to snowflakes are only crude approximations to the complex real processes. Thus, the snowfall produced in the models is substantially less validated against observations and therefore still rather uncertain.

**Table 1.** List of the models analyzed in this study

| Models used | Grid name | Resolution at 0N | Resolution at 50 N | Atmosphere | References |
|---|---|---|---|---|---|
| HadGEM3-GC31-HM | N512L85 | ~40 km | ~25 km | MetUM-GA7.1 | Williams et al. (2017) |
| HadGEM3-GC31-MM | N216L85 | ~90 km | ~60 km | MetUM-GA7.1 | Williams et al. (2017) |
| EC-Earth3-HR | T511L91 | ~40km | ~35km | IFS CY36r4 | Haarsma et al. (2018) |
| EC-Earth3 | T255L91 | ~80km | ~70km | IFS CY36r4 | Haarsma et al. (2018) |
| MPIESM-XR | T255L95 | ~50 km | ~35 km | ECHAM6.3 | Stevens et al. (2013) |
| MPIESM-HR | T127L95 | ~100 km | ~65 km | ECHAM6.3 | Stevens et al. (2013) |
| ECMWF-HR | Tco399L91 | ~25 km | ~25 km | IFS CY43r1 | Roberts et al. (2018) |
| ECMWF-LR | Tco199L91 | ~50 km | ~50 km | IFS CY43r1 | Roberts et al. (2018) |

## 2.2 CloudSat snowfall retrievals

Launched in June 2006, nearly a decade long data of snowfall estimates are derived from the active Cloud Profiling Radar (94 GHz) onboard NASA's CloudSat satellite. While primarily designed for studying the cloud vertical structure, CPR/CloudSat has proved immensely useful in providing precipitation estimates globally (cf. Stephens et al. (2018) for an overview). The radar has an intrinsic vertical resolution of 485 m, but measurements are oversampled to yield profiles at an effective vertical resolution of 239 m. CloudSat observes falling snow between $82^oN$ and $82^oS$ latitude along a ground track with a repeat cycle of 16 days (Kulie et al., 2016; McIlhattan et al., 2017). Due to its sun-synchronous orbital configuration, the sampling is better at high latitude regions (especially around $70^oN$) thus providing the first near global estimates of snowfall (Kulie et al., 2016; Hiley et al., 2011; Kulie and Bennartz, 2009). In the present study, the 2C-SNOW-PROFILE product (Version 5.0) that gives the snowfall accumulation in mm/month from 2006 to 2016 is used for evaluations (Wood, 2011; Wood et al., 2013). CloudSat observations are not available from May through October in 2011 due to a battery failure. There were two other brief anomalies due to battery failures in December 2009 and January 2011, however, neither lasted the whole month so we still have data for those months for the analysis.

Uncertainties in the CloudSat snowfall estimates derive from numerous sources including the need to assume an exponential particle size distribution with temperature-dependent number concentration, the lack of explicit information about particle density, potential influences of attenuation from supercooled liquid water, and the blind zone induced by ground clutter contamination in the four lowest CPR range bins that extend to 1 km above the surface (Hiley et al., 2011; Kulie and Bennartz, 2009). The impacts of these uncertainties have been assessed through numerous prior studies that compare CloudSat snowfall esti-

mates to ground-based radar, in situ accumulation measurements, and seasonal and continental-scale accumulation estimates from reanalyses. While each source of snowfall information used in these studies has its own strengths and weaknesses precluding absolute error estimates from being derived, these studies generally suggest that the CloudSat snowfall product performs well over mid- and high-latitude regions. Comparisons against ground-based radar networks in the United States and Sweden,

for example, suggest that CloudSat reproduces snowfall frequency and accumulation with to within 25% of ground-based radar over the range of scenes where the latter provide (Smalley et al., 2014; Norin et al., 2015, 2017). Palerme et al. (2014, 2017) further demonstrate that on continental scales, CloudSat reproduces seasonal snowfall accumulations in the ECMWF Interim reanalyses with high fidelity. While ERA-interim regional snowfall accumulations suffer from model uncertainties, the integrated accumulation over Antarctica ultimately represents the net water vapor convergence over the ice sheet. Since

the reanalyses routinely assimilate water vapor from satellite observations, this integrated accumulation is well-constrained by independent satellite observations and provides a strong constraint on the net snowfall over the ice sheet. This result is further supported by Boening et al. (2012) who show remarkable consistency between estimates of recent Antarctic snowfall variability derived from reanalyses and CloudSat and completely independent ice sheet mass estimates from the Gravity Recovery and Climate Explorer (GRACE) satellite.

Never-the-less, a number of recent studies have pointed out the inherent limitations in the CloudSat observations that must be acknowledged when considering the results that follow. For example, due to contamination from ground clutter, CloudSat snowfall estimates must be extrapolated from 1 km above the surface (Smalley et al., 2014). This has implications for the snowfall estimates in those regions in the Arctic where low level supercooled liquid clouds or diamond dust that precipitate very light snow are observed (Lemonnier and Wood, 2019). The snowfall from these systems could be either underestimated

or missed entirely by CloudSat (Bennartz et al., 2019) although a recent study by Maahn et al. (2014) showed that a fraction of this under-estimate may be offset by snow virga that CloudSat also fails to represent below 1 km. It is beyond the scope of the current study to add to the existing body of literature concerning the evaluation of CloudSat snowfall estimates but additional discussion of the strengths and limitations of the dataset can be found in Panegrossi et al. (2017); Milani et al. (2018).

Another important limitation associated with the CloudSat snowfall observations is the limited spatial sampling provided by
the nadir sampling CPR. This is somewhat mitigated at high latitudes where sampling is greatly increased over lower latitudes, in particular if data are regridded to coarser resolution.

## 2.3 Methodology

The models output snowfall as snowfall flux ($kg/m^2/s$). This monthly flux is converted into snowfall rate in mm/month. For the analysis, the model output is re-gridded to a $1x1^o$ grid. However, in the case of the observations, CloudSat has poor sampling
(as mentioned in Section 2.2) along the latitudes as the spacing between the longitudes in the Arctic is very low. Hence, so as to avoid patchiness, CloudSat data are accumulated over a 3 degrees longitude and 1 degree latitude grid and averaged over the seasons. This would still provide a sufficient number of samples to compute robust statistics. The following subsections describe the methodology adopted in this study to evaluate the model-derived snowfall against observations.

### 2.3.1 Analysis of snowfall percentiles

The seasonality in monthly accumulated snowfall amounts over the Arctic (north of $50^{o}$N) is evaluated in the model simulations against CloudSat retrievals. In this study, the results are presented for the autumn (Sept-Oct-Nov), winter (Dec-Jan-Feb) and spring (Mar-Apr-May) seasons separately. Since the snowfall distribution is often skewed, we evaluated percentile thresholds rather than averages to properly take into account the spread in the snowfall distribution. Three percentile thresholds, p10, p50 and p90 are used. The evaluation of p10 and p90 provides information on how the light and extreme snowfall events are captured by models respectively and p50 corresponds to the median snowfall. Since the CloudSat observations span the latest 10 year period from 2006 to 2016, a similar ten year period from 2005 to 2014 is chosen from the model simulations.

### 2.3.2 AO analysis

The AO is the most dominant mode of natural atmospheric variability in the Arctic (Thompson and Wallace, 1998, 2000). For the observational reference, AO index from the NOAA webpage (https://www.ncdc.noaa.gov/teleconnections/ao/) is chosen. Since the CloudSat data spans only 10 years, the snowfall accumulation associated with all the positive and negative AO cases are considered. The model simulated AO index (Thompson and Wallace, 1998) is defined as the first leading mode of Empirical Orthogonal Function (EOF) analysis of monthly sea-level pressure anomalies poleward of $20^{o}$N latitude. The extended time period from 1980-2014 is used here to compute the AO index so as to disregard the internal/interannual variability arising from other sources.

## 3 Results

### 3.1 Statistical evaluations using CloudSat retrievals

Figs. 1-3 respectively show the p10, p50 and p90 percentiles in snowfall from CloudSat (first row) and from the Hi-res models (rows 2-5). The three columns show the comparison for three seasons. The snowfall estimates from the Lo-res models are shown in the supplementary (Figs. S1-S3). To establish that there are a sufficient number of snowfall events to calculate the statistics, the total number of snowing pixels available at the original 1x1 deg CloudSat grid, accumulated for each season studied here from 2006 to 2014, and the monthly time series of the number of snowing pixels accumulated over the three selected regions shown in Fig. 5 are presented in the supplementary (Fig. S4). It can be seen that the monthly averages during the SON, DJF och MAM months are not represented by just a few strong events and there are more than 2000 snowing pixels, depending on the region and the season.

The models simulate the spatial distribution of light snowfall (Fig. 1) reasonably well over southern Greenland, Eurasian Arctic and north western Pacific during the winter months (DJF). However, p10 thresholds are up to a factor of 3 higher in the majority of the Hi-res models compared to the CloudSat observations over these regions, suggesting heavy overestimation of moderate snowfall and a strong negative skewness in the snowfall distribution in the models compared to the CloudSat observations. Here, it needs to be noted that CloudSat misses or underestimates the light snowfall below 1000 m due to

contamination from ground clutter as explained in Section 2.2. This means that when analyzing the percentiles, the p10 in CloudSat could further shift to the lower values if we take into account these light snowfall events. This could result in an even larger difference between models and satellite retrievals. However, the knowledge about the frequency of occurrence of such events is lacking as we do not have sufficient observations covering the Arctic.

Among the Hi-res models, MPI-ESM captures the light snowfall over these regions reasonably well. Excessive light snowfall is simulated over eastern Europe and Russia in winter compared to the observations. The Hi-res models, in general, tend to overestimate light snowfall in those high orographic regions such as along the Rocky and Ural mountain ranges and also at the border between the west Siberian plains and the Central Siberian upland during winter, which is not observed in the CloudSat retrievals. During autumn and spring months, a similar tendency is seen, but, the models simulate the light snowfall well over

Eurasia. The 10 percentiles lie around 5-10 mm/month in the Hi-res models, whereas this threshold is around 2-4 mm/month in the observations during these months. The southward extend of light snowfall that is observed in CloudSat, particularly over Europe/western Eurasia and over northern north America is not simulated by the models. Irrespective of the season, the models strongly underestimate the light snowfall over the Gulf of Alaska.

     The median snowfall (Fig.2) is represented spatially well by the models, including over the mountainous regions in all the

seasons. The regridding applied to the CloudSat snowfall data to obtain sufficient samples and to increase the robustness of the results, has smoothened out such specifics to a certain extent compared to the Hi-res simulations. The increase of snowfall over the regions of high orography is still evident during the winter months. Here too, the snowfall distribution in the models is negatively skewed compared to the observations, with the median snowfall overestimated by up to a factor of 1.5 in the Hi-res model set ups in winter. The median snowfall lies around 40-50 mm/month in the heavy snowfall regions in the observations

such as over southern Greenland and over the Andes mountain range, whereas in the models this lies above 60 mm/month. The models simulate the snowfall reasonably well during autumn and spring months both spatially and in magnitude, however, it has to be noted that the median snowfall in the models do not extend as far as $50^oN$ as is in the observations.

     The accumulated extreme snowfall amounts, expressed as the $90^{th}$ percentile is shown in Fig. 3 from observations and Hi-res model set ups. In this case, an opposite picture to that seen in the light and median snowfall is evident. Here, the models

do not capture the spatial distribution of extreme snowfall, particularly, in autumn and spring months. The snowfall amount is markedly underestimated in all the seasons by all the models. In winter, though the models simulate the regions of heavy snowfall realistically over south Greenland, over western Pacific, they underestimate the magnitude by up to almost one-half compared to the observations. In this case, the snowfall distribution simulated by the models are positively skewed with the $90^{th}$ percentile value lying between 100-150 mm/month in the CloudSat retrievals and between 50-100 mm/month in the Hi-res

model simulations.

     These results are consistent with the study by Kay et al. (2018) applying a CloudSat simulator in the Community Earth System Model (CESM version 1) to evaluate the precipitation globally. Fully coupled simulations also showed similar tendencies, such as excessive light snow and inadequate heavy snowfall amounts as atmospheric only simulations over mid- and high latitudes. This means that the main biases are from the atmospheric model due to the simplified parameterizations used in the

representation of the complex ice processes.

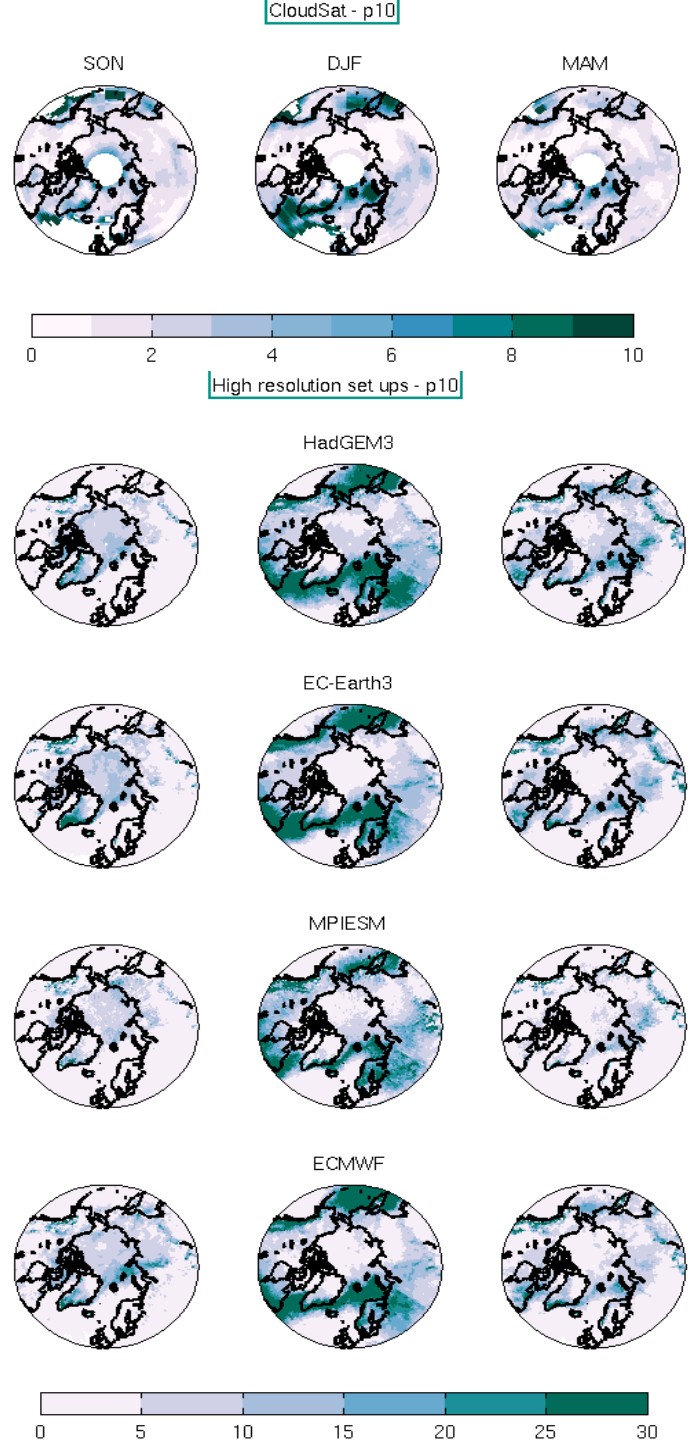

**Figure 1.** 10-percentile thresholds of monthly snowfall accumulations (mm/month) for the SON, DJF and MAM months in the 3 columns respectively. The top row shows the CloudSat observations and the other four rows below show snowfall from the Hi-res set ups of HadGEM3, EC-Earth3, MPI-ESM and ECMWF respectively.

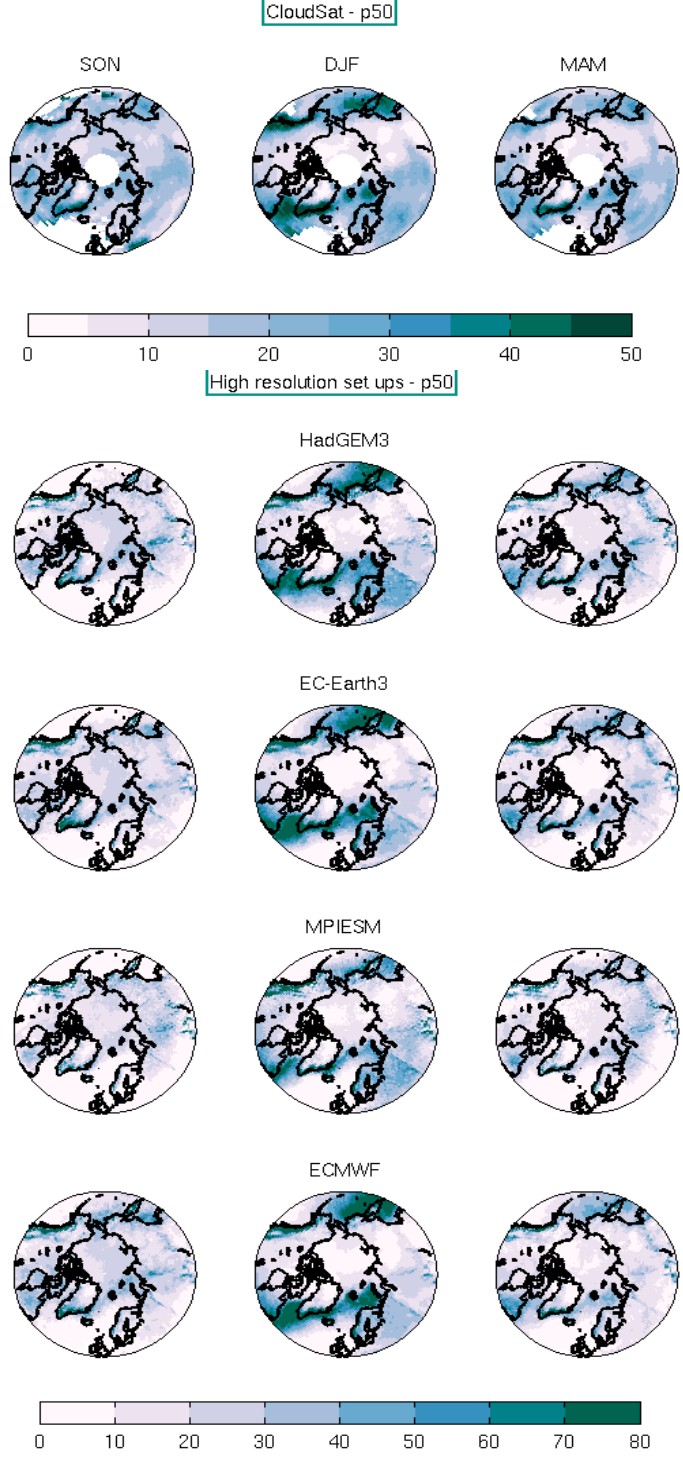

**Figure 2.** Same as Fig. 1, but, at 50-percentile threshold.

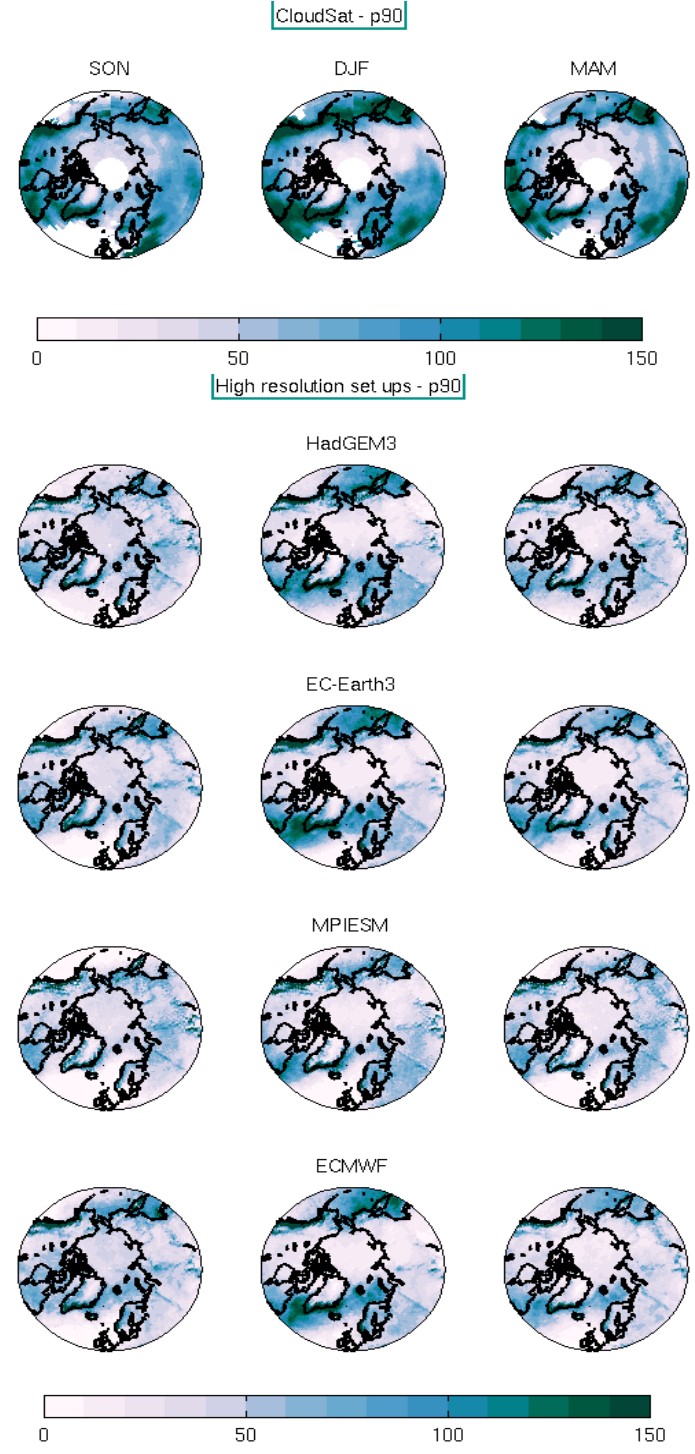

**Figure 3.** Same as Fig. 1, but, at 90-percentile threshold.

## 4 The impact of higher spatial resolution

As discussed in the previous section, the Hi-res model set ups tend to overestimate light snowfall and underestimate extreme snowfall amounts more strongly in winter compared to the other months. To understand if increasing the spatial resolution would impact the snowfall distribution, the difference between the Hi-res model set ups and their Std-res counterparts is analyzed and shown in Fig. 4 for the winter months. In this section, we focus on three main regions, namely, north Pacific, Eurasian Arctic ocean and south of Greenland. The columns 1-3 denote the three percentiles, p10, p50 and p90 respectively.

Over the north east Pacific, the Hi-res set up of ECMWF overestimate light snowfall (i.e. 10 percentiles) by around 10-15 mm, whereas the Hi-res HadGEM3 model underestimates this compared to its Std-res counterpart. This means that the Hi-res set up of HadGEM3 model reduces the positive bias, whereas the Std-res set up of ECMWF model reduce the positive bias over this region. The Hi-res ECMWF and EC-Earth model versions overestimate the light snowfall amounts over Norwegian and Greenland Seas. Light snowfall is overestimated in the Hi-res HadGEM3 model over south of Greenland, whereas this is underestimated in EC-Earth and ECMWF models. Light snowfall is underestimated in the Hi-res ECMWF model along the east coast of Greenland. A change in resolution do not impact the simulation of light snowfall in the MPI-ESM model over these regions.

No notable change in the simulation of median snowfall (i.e. 50 percentiles) with change in resolution is seen in all the three models over the north Pacific, except HadGEM3 wherein a strong underestimation is seen in its Hi-res version. There is a marginal overestimation in the Barents Sea by the Hi-res EC-Earth model and an underestimation south of Greenland. A very patchy picture is seen in the median snowfall differences between the Hi-res and the Std-res set up of MPIESM where no clear conclusions can be drawn. Over southern Greenland, the Hi-res set up of HadGEM3 overestimate the snowfall amount whereas an opposite sign is seen in the EC-Earth model.

As explained in the previous section, the Hi-res set ups of all the models used in this study underestimates the extreme snowfall (90 percentiles) by more than 50%. Striking changes can be seen in the all the models with increasing resolution in this case. The Hi-res set up of HadGEM3, EC-Earth and ECMWF models overestimate the extreme snowfall over the Norwegian, Barents and Greenland Seas compared to their low resolution counterparts. This means that the Hi-res set up of these models improves the simulation of extreme snowfall in these regions. Similarly, an improvement can be seen around the Bering Strait in HadGEM3 and ECMWF models. The differences are patchy in the MPIESM model.

It is previously reported that the impact of using a higher model resolution is more profound when going from a coarser than $1^o$ to about 50 km grid resolution, but only have relatively small changes for further resolution increases to 10-20 km (Jung et al., 2012) in the simulation of extratropical cyclone characteristics. In this study, the high-latitude model resolutions vary from 50-60 km to ~25 km. Therefore, it is not surprising that only marginal changes are found. However, it needs to be noted that the resolution impact for single snowfall events on smaller temporal scales might differ from the monthly accumulated snowfalls.

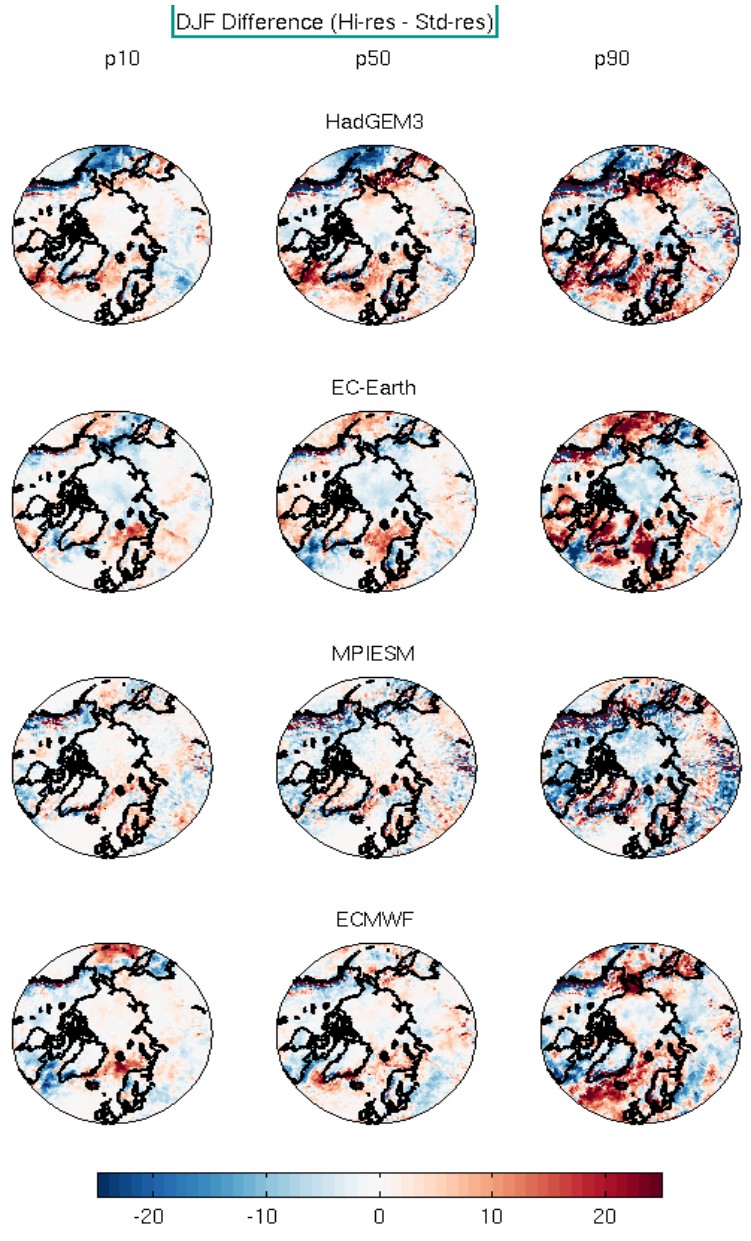

**Figure 4.** The differences between Hi-res and Std-res model simulated snowfall (Hi-res minus Std-res) for the DJF months for the 10 (left column), 50 (center) and 90 percentiles (right column).

## 5    Seasonality and interannual variability over selected regions

To analyse the interannual variability in snowfall, four regions are selected, three as shown in Fig. 5 and the Arctic region, north of 70N. The selection is based on the Figs. 1 to 3. These regions show high snowfall variability. Regions 1 to 3 are southern Greenland, Eurasian Arctic ocean and north-west Pacific respectively. The time series of average snowfall in mm/month is presented in Fig. 6 for these three regions for the period 2005-2015. CloudSat observations are not available from May through October in 2011 due to a battery failure. The ensemble mean of monthly accumulated snowfall amounts from the Hi-res models and Std-res models are presented as the red and green curves and the CloudSat retrievals as the blue curve. Over southern Greenland, north-west Pacific and Arctic, the simulated snowfall is overestimated, irrespective of the model resolution. Similar over-estimation in snowfall is also simulated in the CMIP5 models over Antarctica (Palerme et al., 2017). The models seem to agree well with the observations over the Eurasian Arctic ocean. It can be noted that the wintertime seasonality in snowfall is more prolonged in the models, irrespective of the region, compared to the CloudSat observations. Simulation of snowfall is almost insensitive to the change in resolution over these regions.

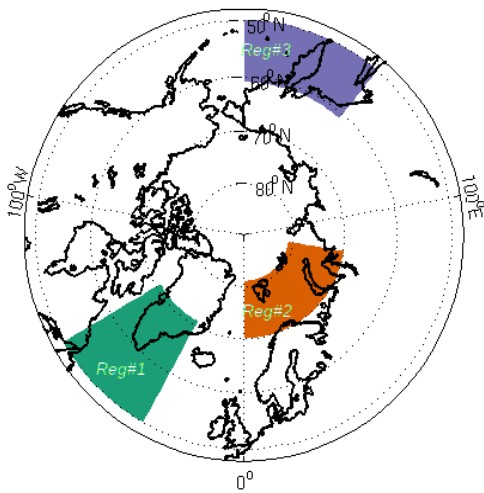

**Figure 5.** Regions selected for this study. Reg1: Southern Greenland Reg2: Eurasian Arctic ocean and Reg3: North West Pacific.

The observed differences between the models and CloudSat observations can be best explained by investigating the statistical distribution of snowfall accumulation over the selected regions. Figure 7 shows this comparison of these distributions. In addition to the selected three regions, the snowfall distribution covering nearly the entire Arctic (70N-82N, 180W-180E, denoted as R4) is also shown. It is evident that the snowfall follows an exponential distribution, while all models show a Gaussian distribution that is heavily positively skewed. The light snowfall amounts are strongly under-represented in the models. The right hand tail of the distribution in CPR/CloudSat derived snowfall is much longer compared to the models. It is also interesting to note that the distributions have different shapes over the three selected areas, both in the models and observations, suggesting the importance in evaluating the distributions regionally.

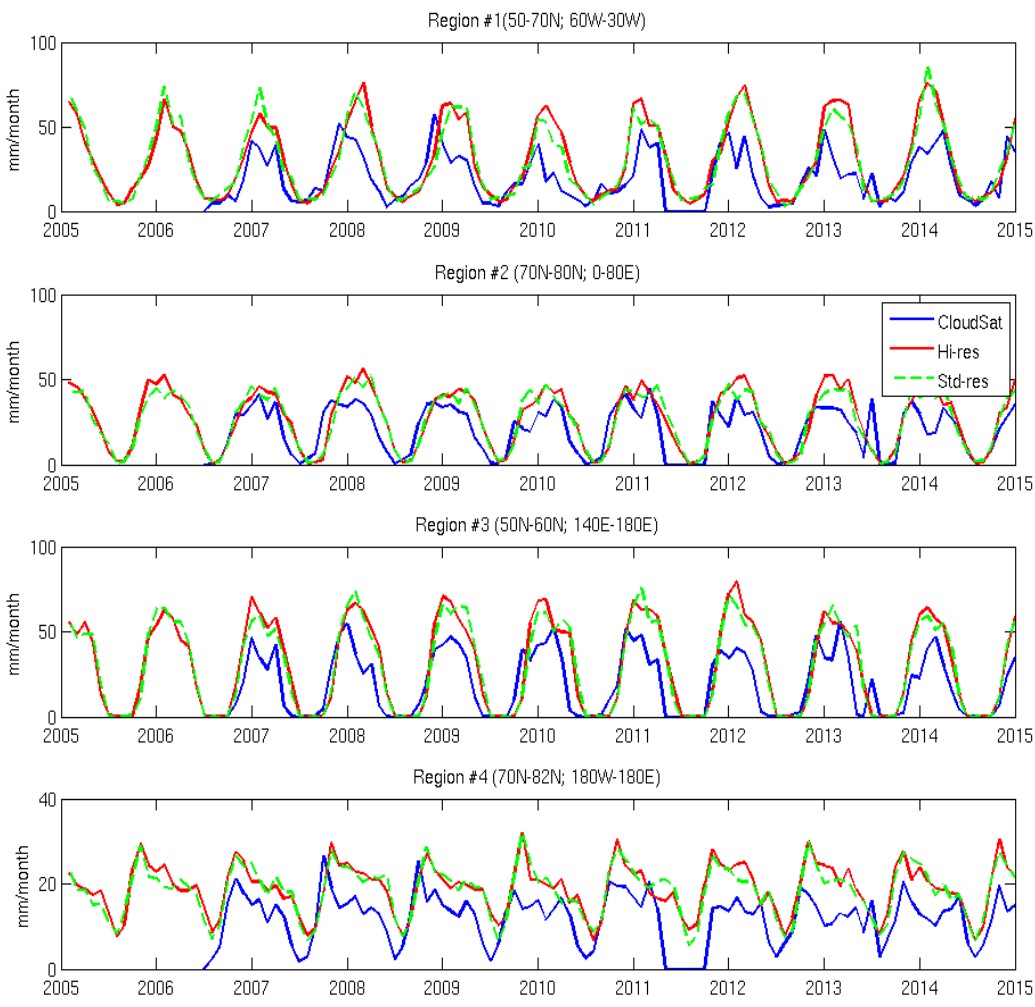

**Figure 6.** Time series of snowfall (mm/month) for the period 2005-2015 from CloudSat (blue line), Hi-res ensemble model mean (red) and Std-res ensemble model mean (green) over the three regions shown in Fig. 5 and over the whole of Arctic.

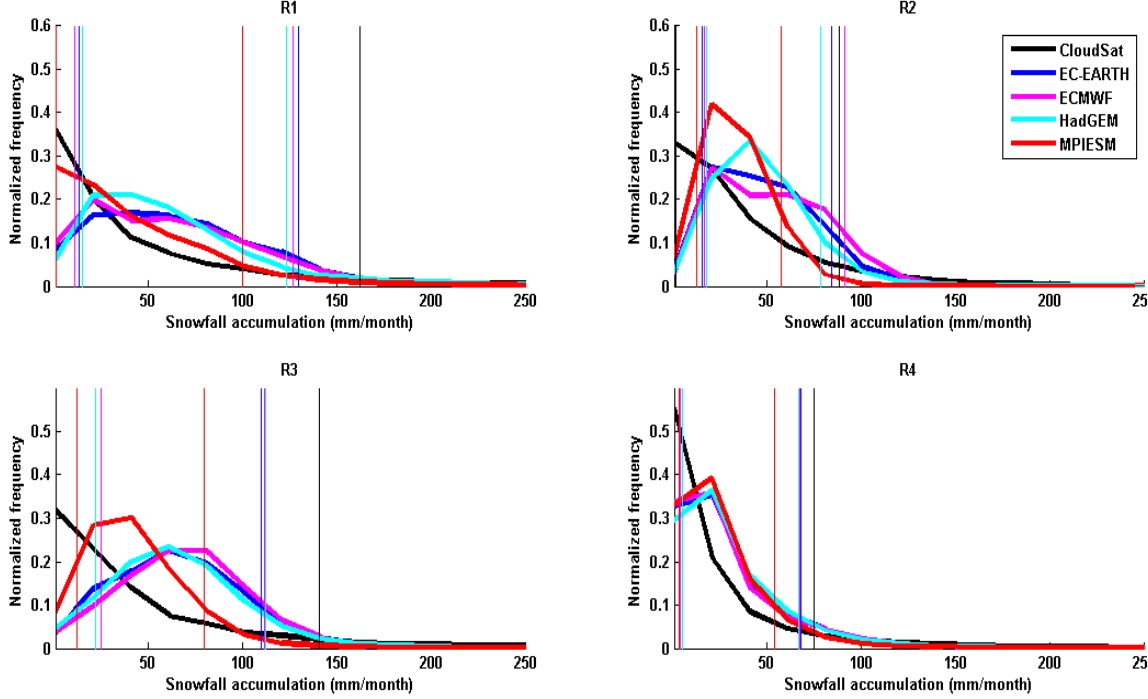

**Figure 7.** Statistical distribution of snowfall accumulation (mm/month) over the three regions (R1-R3) shown in Fig. 5 and R4: over entire Arctic (70-82N) from Hi-res models and CloudSat observations. The vertical lines denote the p10 and p90 percentiles.

To summarize the model evaluation in a more quantitative manner and to add another dimension to the evaluation, the standard deviations, root mean squared differences and Pearson's correlation coefficients are presented in Fig. 8 as a Taylor diagram using the entire time series for the three regions shown in Fig. 5 and as well as for the Arctic region (70N-83N, 180W-180E, denoted as Reg4). The colour-filled circles show results for the high resolution versions and the empty circles their standard resolution counterparts. The correlations typically range between 0.6-0.8, Over all the regions and in the models, the snowfall variability is higher compared to the CloudSat observations. The regional differences among models are strong. For example, over Reg1, the standard deviations have large spread among models, while over Reg3 the models tend to cluster together (except MPIESM versions which are closest to the observations) and have similar variability. The comparison of standard resolution model versions with their high resolution counterparts against CloudSat observations does not show a clear improvement in the high resolution versions or a particular tendency that holds across all models.

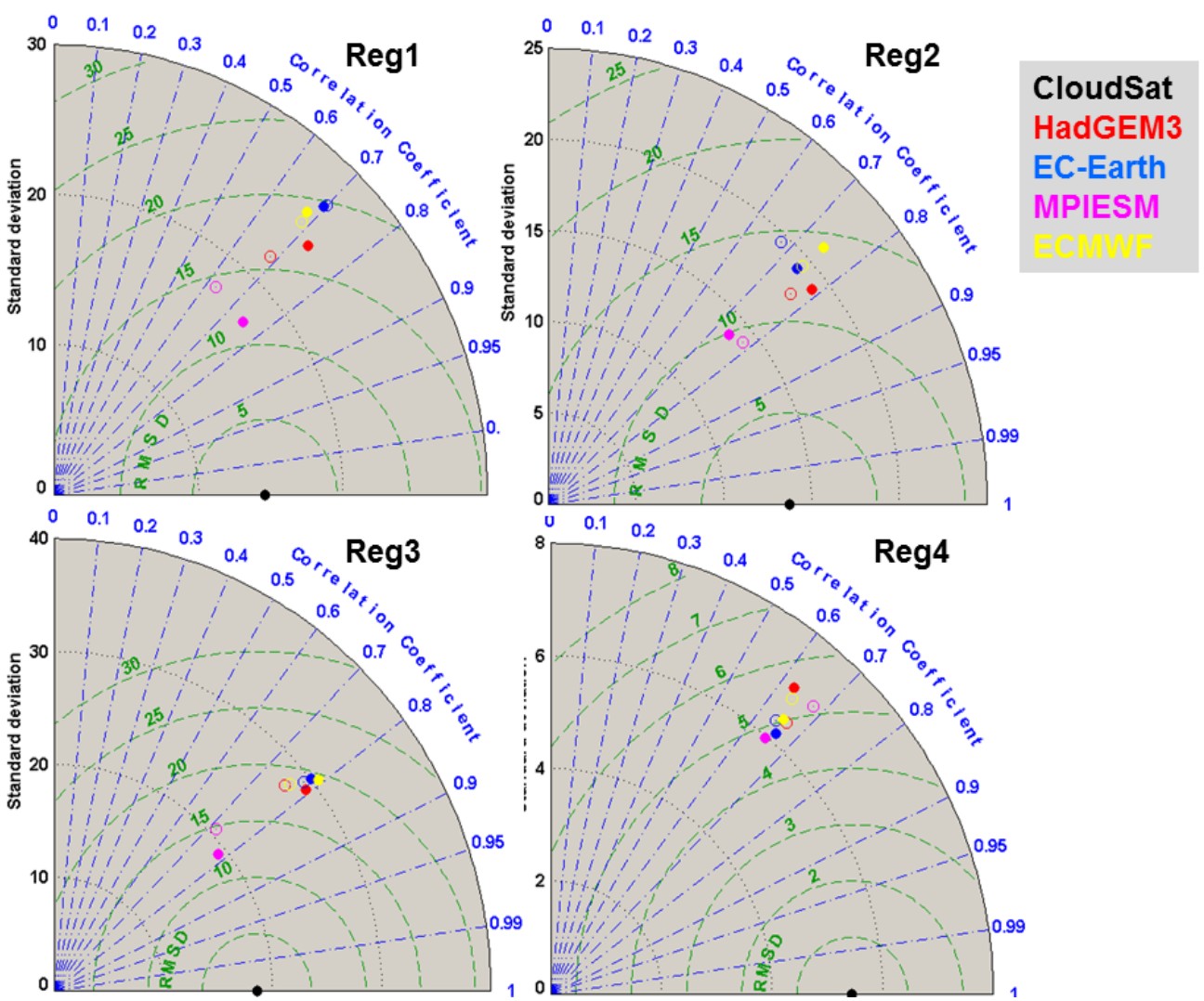

**Figure 8.** Taylor diagram of snowfall accumulation over the four regions from both Hi-res and Std-res models with respect to CloudSat observations.

## 6 Response of snowfall to the AO

AO determines the degree to which the Arctic air penetrates into the mid latitudes and vice versa. The phases of the AO determine the interannual precipitation variability not only in the Arctic, but also over Eurasia and North America (Bamzai, 2003). Considering the importance of wintertime snowfall variability associated with the AO in the Arctic climate system, we
evaluated how well the models in question are able to capture the changes in the snowfall associated with the positive and negative phases of winter AO (DJF).

Fig. 9 shows the snowfall response to the AO in terms of the differences in snowfall amounts between the positive and negative phases of the AO. The top row shows this response in the CloudSat observations. The other rows show the same for the Hi-res and and Std-res set ups. During the positive phases, the cyclonic systems penetrate deeper, northward into the
Greenland Sea and central Eurasian Arctic, leading to increased snowfall over these regions. In these cases, the snowfall is reduced over southern Greenland (Appenzeller et al., 1998). This characteristic snowfall response to the AO is captured well in the CloudSat observations. All models however show a consistence increase in the oceanic snowfall from the northern North Atlantic to the northernmost parts of the Greenland Sea. Irrespective of the models or model resolutions, the observed decrease in snowfall over southern Greenland is not simulated. This is due to the fact that the North Atlantic storm tracks during the
positive phase of AO are either too zonal or shifted more southward in the climate models resulting in weaker cyclonic systems over the Greenland and Norwegian sea and stronger systems over continental Europe (Zappa et al., 2013). The Hi-res versions of the EC-EARTH and ECMWF models show even a stronger snowfall response to the AO over the oceanic regions around Greenland compared to their Std-res counterparts. The opposite tendency is observed in the case of HadGEM and MPI-ESM models over the same region. The CloudSat observations further show increased snowfall over continental regions covering
Ural mountains and West Siberian Plains. All models also show this increase, albeit to a varying degree. In the Pacific sector of the Arctic, CloudSat observations do not show any clear, robust snowfall response to the AO. The simulated snowfall response in the Hi-Res and Std-res set ups of EC-Earth is strikingly different from one another in this sector.

**Figure 9.** Difference between positive and negative phases of AO (AOP-AON) in snowfall accumulation (mm/month) from CloudSat (top row) and models (rows 2-5). The Hi-res set ups are on the left and their Std-res counterparts are on the right.

## 7 Conclusions

The ice processes and their parameterizations in the climate models, especially snowfall, are not adequately evaluated using observations. Given the importance of snowfall both from the climate and weather perspectives and the recent availability of snowfall estimates from CloudSat, we carried out a detailed comparison from a wide range of climate models at two horizontal resolutions each with satellite observations, with focus over the Arctic. The following conclusions can be drawn from the comparisons.

The statistical distribution of snowfall is narrower in the GCMs compared to the CloudSat observations. In the case of light snowfall (10 percentiles), all high resolution versions of the GCMs that are investigated here simulate the spatial distribution of light snowfall realistically. However, the 10th percentiles are overestimated by a factor of up to 3 in all the GCMs compared to the CloudSat observations. The Hi-res model set ups overestimate light snowfall over the mountainous regions such as along the Rockies and Ural mountains compared to their Std-res counterparts. The median snowfall represented by 50 percentiles is also up to a factor of 1.5 high in the models. The median snowfall lies around 40-50 mm/month in those heavy snowfall regions such as over southern Greenland and over the Andes in the observations, whereas these percentiles lie above 60 mm/month in the models. While the extreme snowfall accumulation (90 percentiles) is simulated better by the models, they are, in contrast to light and median snowfall, underestimated compared to the CloudSat observations.

The main reason behind the observed differences in the 10, 50 and 90 percentiles of monthly snowfall accumulation is the fact that the CloudSat and models have different statistical distributions. The observed snowfall distribution follows an exponential distribution over the Arctic (north of $50^o$ N), while all models follow a Gaussian distribution that is strongly positively skewed. This indicates that the light drizzle in the models is not adequately sensitive to the triggering/initiation and sustenance processes. On the other side of the statistical distribution, the models often remove cloud water quicker than expected in the heavy precipitation scenario, thus not allowing the building up of extreme snowfall as observed by CloudSat.

The wintertime seasonality in snowfall is more prolonged in the models compared to the CloudSat observations. The overestimation in model simulated snowfall in p10 and p50 percentiles is strongest during the DJF months. This indicates that the hydrometeor phase partitioning in the models is probably not realistic, in that the supercooled cloud liquid water and light liquid drizzle are underrepresented. This has implication for the local radiation budget as the dynamical and radiative impacts of having prolonged wintertime snowfall at the expense of adequate liquid precipitation can be quite different.

Apart from the traditional statistical comparisons summarized above, we further investigated the snowfall response during the Arctic Oscillation. Such process-oriented evaluation provides an additional insight into how the models simulate a process that is a key for representing the dominant natural variability over this region. The characteristic snowfall variability during the AO, with increased snowfall over Greenland Sea and central Eurasian Arctic and reduced snowfall over southern Greenland and continental Europe is captured well in CloudSat despite the short time period of the observations. The models simulate the increased snowfall in the above mentioned regions realistically at varying magnitudes, but, the snowfall reduction over southern Greenland is not simulated by any of the models.

Finally, since one of the main aims of the PRIMAVERA project is to examine the importance of having high spatial resolution, here, the high resolution model simulations are contrasted against their standard resolution counterparts. Although some regional improvements are seen in the snowfall estimates with a change in atmospheric resolution within a particular model, these improvements are minor and it is not easy to single out a particular pattern or systematic behavior that holds across all
high resolution models. This indicates that representing physical processes accurately in models is more important than purely improving spatial resolution, although both go hand in hand to a certain extent.

In a cautionary note, it should be acknowledged here that, although 10 years of snowfall estimates are now available from CPR/CloudSat, this time period is still shorter considering that the natural/internal variability can occur on multi-decadal time scales. It is therefore not expected that the models simulate all regional features realistically compared to CloudSat
observations. CloudSat nonetheless provides the most reliable estimates of snowfall to date globally and hence such evaluation provides insight into how well models can simulate snowfall to a first order.

*Code and data availability.*  Access to the model output data used in this study will be available through the European Research Council Horizon 2020 PRIMAVERA project (https://www.primavera-h2020.eu/modelling/data-access/). More information regarding model configurations and data availability are available from the authors upon request. All CloudSat data used here are freely available through the
CloudSat Data Processing Center and at the time of writing can be accessed online at http://www.cloudsat.cira.colostate.edu. The matlab and cdo scripts used in this intercomparison are available from the lead author upon request.

*Author contributions.*  The first author performed the analysis and drafted the manuscript. Tristan L'Ecuyer provided the snowfall estimates from CloudSat. All the co-authors contributed to the interpretation of the results and reviewing the manuscript.

*Acknowledgements.*  This study was financially supported by PRIMAVERA (PRocess-based climate sIMulation: AdVances in high resolution
modelling and European climate Risk Assessment), a Horizon 2020 project funded by the European Commission.

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
