# Peer review of "Snowfall distribution and its response to the Arctic Oscillation: An evaluation of HighResMIP models in the Arctic using CPR/CloudSat observations"

_Geoscientific Model Development, 2019_

## Referee Comment (RC1) · Anonymous Referee #1 · 15 Apr 2019

This article presents a comparison of simulated snow accumulation from high resolution model simulations contributing to HighResMIP, and satellite estimates of snowfall based on retrievals from CloudSat-CPR. The topic is interesting, and the comparison itself is novel, with respect to the evaluation of snow simulation in this new set of high-resolution modelling experiments. However, I have some concerns about the description of the methods, and some of the interpretation, which are described below.

1. Inadequate methodological description. I found that the methods used were not adequately described. For instance, how was the regridding of CloudSat measurements

to 1 x 3 degrees performed? Were all instantaneous profiles within a grid box simply averaged together, or something else? On P3 L25 it states that model data were re-gridded to 1x1 degree, so how were comparisons made between model and CloudSat if the grids are different? Further, P4 L18 states that 1x3-degrees provides "sufficient" sample size, but how was "sufficient" determined, and what constitutes "robust"? Was the comparison of mm snow water equivalent (SWE), or snow depth? If the former, then the conversion method and treatment of snow density should be included.

Another concern is the treatment of missing data: on P12 L5 the 2011 battery failure is discussed; however, we are aware of two other battery failures [September 2009 - December 2009] and [January 2011], but these periods are seemingly unaffected in Fig.6. Was any gap-filling/interpolation required during these times?

2. Set of models used for comparison. When the authors mentioned an evaluation of low, versus high, resolution models, I expected to see at least a couple of "typical" CMIP5-class GCMs/ESMs included, to provide a reference for how/if these high resolution simulations represent an improvement for the simulation of snow. The low/high res sample evaluated here is somewhat artificial, since even the "low res" models are among the highest resolution simulations one would find in CMIP5. I would strongly recommend that the authors include one, or more, CMIP5-era simulations, to provide some more context for the HighResMIP results.

3. Figure quality. I found the maps very difficult to read, because of the small size and choice of colour scale. In general, it is very challenging to ask readers to evaluate by eye how well a set of model simulations compares to a reference map. I strongly recommend condensing these maps into standard model v obs diagnostics, for instance using Taylor Diagrams. This would provide a much more rigorous evaluation of each model than can be provided by eye, and would make it easier for the authors to add more models to the comparison (see point 2 above). In addition, the CloudSat map panels have a data-wrapping issue at the date line.

4. Uncertainties. The devil is in the details with this type of comparison, and I would have appreciated a much more thorough discussion of the various sources of uncertainty that are present in these results. A much-needed addition to Fig.6 could be a credible interval for each estimate, considering sampling, instrumental and model uncertainty. Due to sampling, I would expect to see much larger uncertainties on the CloudSat snow estimates coming from more southerly latitudes (e.g. Reg#3).
* * *

---

## Referee Comment (RC2) · Anonymous Referee #2 · 6 May 2019

This paper presents an evaluation of a few climate models within the HiResMIP project in their representation of Arctic snowfall. The authors show that the models show significant biases compared to the CloudSat derived snowfall rates, and that there is no clear improvement in the higher-resolution models compared to low-resolution models. While potentially interesting, this paper lacks a lot of details on the observational uncertainty, model choice, statistical robustness, results analysis, and discussion. The authors are invited to expand on the initial analysis and consider the points below in that process. I focus on the main points first, after which I highlight some more minor

points.

Major comments

1. The observational uncertainty is very poorly quantified. CloudSat suffers from main issues with undersampling light precipitation events (as the authors indicate), but no uncertainty is assigned to that. I wonder why the authors did not consider working with other CloudSat data (including internal quality flags of the CloudSat algorithm, and/or radar-derived reflectivity).

2. Statistical robustness and process analysis. If working with percentiles, while only considering <10 years of data (so only a sample of <30 months for each season), I wonder how significant these percentiles are. In addition, the authors should be extremely careful in translating a low observed monthly snowfall rate to an inadequate treatment of light snowfall by the models; it might very well be that this monthly snowfall rate comes from just one large snowfall event! Although I realize that CloudSat frequency is only monthly for this gridded product, this connection from observation to model physical processes should be much more expanded upon; for example, the authors can look into the distribution of snowfall in the models in those low-snowfall months, to confirm whether or not these are associated with just low-snowfall events or not. Without such a detailed analysis, the reader remains 'in the dark' about the exact defiency in the models. Note that recent literature suggests that models produce 'too little, too frequent' snowfall (e.g. https://journals.ametsoc.org/doi/pdf/10.1175/JCLI-D-16-0666.1); does that apply do these models as well?

3. Choice of models: why were only these models chosen? Why only ocean-forced models and not fully-coupled models? Why is opted to not use a COSP-style radar simulator in the models? How does that impact the results? More details on the PRIMAVERA project are necessary. In any case, if only AMIP-style models are chosen, the authors should use the overlapping period between observations and models only.

4. Methods, results, and discussion are mixed throughout the paper, which is somewhat confusing for the reader. A conclusion section with a long list reads like notes from a presentation, and should be converted a flowing text instead. More discussion should be added; why are these results important? How do these compare to other literature? What can we learn from it? What processes are lacking and/or inadequate in models?

Minor comments (P=page, L=line)

P1, L2: surface radiation budget

P1, L20: How would incoming radiation cool the surface?

P2, L34: irrelevant for this paper; Alpine snowfall is much more related to mid-latitude atmospheric dynamics, as well as orographic snowfall.

P3, L10: What is this project and what is its aim, and how does this paper fit in that?

P3, L15: what is a snowfall flux? Mass or energy?

P3, L21: later, you discuss results from 2006-2015. Try to be clear

P3, L27: possibly?

P4, L19: how do you end up with ten seasons? Clarify.

P4, L20 to P5, L8: these are methods, not results. Consider changing your section titles to improve structure (Section 2: Data and Methods; Section 3: Results; Section 4: Discussion and Conclusions)

P5, L16: larger difference

P5, L29: extent

P5, L31: what does 'negatively skewed' mean?

P6, L9 and beyond: This belongs to the discussion section

Figures: too small; add units and indicate seasons in Figure itself. Add sublabels (e.g.

a, b, etc) to refer to in text.

P10, L32: this is the main issue of this paper; it should be added to the discussion section and reflected/expanded upon – see Main Issue 2

P12, L5: this is part of Methods, and does not fit here

P15, L1-13: Methods.

Section 6: This has not been clearly introduced in the intro and feels obsolete. Why would this be relevant to this paper? Why not also look at impact of NAO, Arctic sea ice extent, GBI, and other indices?

P17, L4: I would argue that this is definitely not 'a wide range of models'
* * *

---

## Author Comment (AC1) · 12 Jun 2019

**Response to Reviewer #1**

We thank the reviewer for providing constructive remarks. We have tried to incorporate your suggestions in the revised manuscript. Please find below a point by point response to them.

1(a). Inadequate methodological description. I found that the methods used were not adequately described. For instance, how was the regridding of CloudSat measurements to 1 x 3 degrees performed? Were all instantaneous profiles within a grid box simply averaged together, or something else? On P3 L25 it states that model data were regridded to 1x1 degree, so how were comparisons made between model and CloudSat if the grids are different? Further, P4 L18 states that 1x3-degrees provides "sufficient" sample size, but how was "sufficient" determined, and what constitutes "robust"? Was the comparison of mm snow water equivalent (SWE), or snow depth? If the former, then the conversion method and treatment of snow density should be included.

We did in fact started the evaluations on the 1x1 deg grid for CloudSat before settling for the 1x3 deg grid. The latter was chosen in order to avoid having patchiness in the data that arises due to poor sampling along the latitudes as the spacing between the longitudes in the Arctic is very low. A figure below shows CloudSat p50 values over 1x1 deg and 1x3 deg grids respectively. As can be seen the spatial consistency in the snowfall estimates is better in the 1x3 deg grid.

Level-3 gridded snowfall accumulation (mm/month) estimates from CloudSat are used in this study. However, the models provide the snowfall flux (kg/m2/s). This is converted to snowfall accumulation so as to carry out a fair comparison with observed data. These points are now clarified in the revised manuscript.

With regard to the sample size, we kindly refer the reviewer to our response given to Reviewer #2 on the similar topic.

**(a) Original**

[Figure]

**(b) Regridding the data to 1 deg latitude x 3 deg longitude**

[Figure]

(b) Another concern is the treatment of missing data: on P12 L5 the 2011 battery failure is discussed; however, we are aware of two other battery failures [September 2009 - December 2009] and [January 2011], but these periods are seemingly unaffected in Fig.6. Was any gap-filling/interpolation required during these times?

We agree that there were two other brief anomalies due to battery failures in December 2009 and January 2011, but neither lasted the whole month so we still have data for those months to compute the averages. This is now clarified in the revised manuscript.

2. Set of models used for comparison. When the authors mentioned an evaluation of low, versus high, resolution models, I expected to see at least a couple of "typical" CMIP5-class GCMs/ESMs included, to provide a reference for how/if these high resolution simulations represent an improvement for the simulation of snow. The low/high res sample evaluated here is somewhat artificial, since even the "low res" models are among the highest resolution simulations one would find in CMIP5. I would strongly recommend that the authors include one, or more, CMIP5-era simulations, to provide some more context for the HighResMIP results.

We understand that the 'low' resolution models here can be classified as 'high' resolution simulations in the CMIP5 context. The main aim of this project was to understand the effect of

resolution, keeping the model and nudging the same, in simulating the snowfall estimates. A direct and fair comparison cannot be made with the CMIP5 model simulations in this context for the following reasons:

1. The models used in the present study have been improved in terms of physics and parameterization schemes compared to their counterpart versions used in CMIP5 simulations. Hence, it would be difficult to say if an improvement, if seen is indeed due to a change in resolution or change(s) in other processes.
2. Also, the time period of the simulation used here is different than in the CMIP5 simulations.

However, if the reviewer still sees a value in including the results from a typical CMIP5 simulation, we would be happy to include those in the further revision.

3. Figure quality. I found the maps very difficult to read, because of the small size and choice of colour scale. In general, it is very challenging to ask readers to evaluate by eye how well a set of model simulations compares to a reference map. I strongly recommend condensing these maps into standard model v obs diagnostics, for instance using Taylor Diagrams. This would provide a much more rigorous evaluation of each model than can be provided by eye, and would make it easier for the authors to add more models to the comparison (see point 2 above). In addition, the CloudSat map panels have a data-wrapping issue at the date line.

Following the reviewer suggestion, we have now added and described Taylor diagrams in the revised version, as shown below. The standard deviations, root mean squared differences and Pearson's correlation coefficients are plotted using the entire time series for the three regions shown in Fig. 5 in the manuscript and as well as for the Arctic region (70N-83N, 180W-180E, denoted as Reg4). The colour-filled circles show results for the high resolution versions and the empty circles their standard resolution counterparts. The correlations typically range between 0.6-0.8, Over all the regions and in the models, the snowfall variability is higher compared to the CloudSat observations. The regional differences among models are strong. For example, over Reg1, the standard deviations have large spread among models, while over Reg3 the models tend to cluster together (except MPIESM versions which are closest to the observations) and have similar variability. The comparison of standard resolution model versions with their high resolution counterparts against CloudSat observations does not show a clear improvement in the high resolution versions or a particular tendency that holds across all models.

[Figure]

The CloudSat maps are revised to remove the data wrapping at the date line. Thanks for pointing out this plotting artifact. Also the other maps are replotted to improve image quality.

4. Uncertainties. The devil is in the details with this type of comparison, and I would have appreciated a much more thorough discussion of the various sources of uncertainty that are present in these results. A much-needed addition to Fig.6 could be a credible interval for each estimate, considering sampling, instrumental and model uncertainty. Due to sampling, I would expect to see much larger uncertainties on the CloudSat snow estimates coming from more southerly latitudes (e.g. Reg#3).

We have revised and added the following information on the various uncertainties in the CloudSat retrievals and analysis.

[revised manuscript text omitted]

7. Palerme, C., Claud, C., Dufour, A., Genthon, C., Wood, N.B., L'Ecuyer, T.S., 2017. Evaluation of Antarctic snowfall in global meteorological reanalyses. Atmos. Res. 190, 104–112. http://dx.doi.org/10.1016/j.atmosres.2017.02.015.

8. Panegrossi, G., Rysman, J.-F., Casella, D., Marra, A.C., Sanò, P., Kulie, M.S., 2017. CloudSat-based assessment of GPM microwave imager snowfall observation capabilities. Remote Sens. 9, 1263. http://dx.doi.org/10.3390/rs9121263.

9. Milani, L., Mark S. Kulie, Daniele Casella, Stefano Dietrich, Tristan S. L'Ecuyer, Giulia Panegrossia, Federico Porcù, Paolo Sanò, Norman B. Wood, 2018. CloudSat snowfall estimates over Antarctica and the Southern Ocean: An assessment of independent retrieval methodologies and multi-year snowfall analysis, Atmospheric Research 213, 121-135.

10. Bennartz, R., Fell, F., Pettersen, C., Shupe, M. D., and Schuettemeyer, D.: Spatial and temporal variability of snowfall over Greenland from CloudSat observations, Atmos. Chem. Phys. Discuss., https://doi.org/10.5194/acp-2018-1045, in review, 2019.

---

## Author Comment (AC2) · 12 Jun 2019

Norrköping, 2019-06-12

**Response to Reviewer #2**

We thank the reviewer for providing constructive remarks. We have tried to incorporate your suggestions in the revised manuscript. Please find below a point by point response to them.

Major comments:
1. The observational uncertainty is very poorly quantified. CloudSat suffers from main issues with undersampling light precipitation events (as the authors indicate), but no uncertainty is assigned to that. I wonder why the authors did not consider working with other CloudSat data (including internal quality flags of the CloudSat algorithm, and/or radar-derived reflectivity).

We thank the reviewer for this comment, which is also raised by the Reviewer #1. Instead of repeating the same response here, we kindly refer the Reviewer #2 to our response given to the Reviewer #1 on the same topic.

2. Statistical robustness and process analysis. If working with percentiles, while only considering <10 years of data (so only a sample of <30 months for each season), I wonder how significant these percentiles are. In addition, the authors should be extremely careful in translating a low observed monthly snowfall rate to an inadequate treatment of light snowfall by the models; it might very well be that this monthly snowfall rate comes from just one large snowfall event! Although I realize that CloudSat frequency is only monthly for this gridded product, this connection from observation to model physical processes should be much more expanded upon; for example, the
authors can look into the distribution of snowfall in the models in those low-snowfall months, to confirm whether or not these are associated with just low-snowfall events or not. Without such a detailed analysis, the reader remains 'in the dark' about the exact defiency in the models. Note that recent literature suggests that models produce 'too little, too frequent' snowfall (e.g. https://journals.ametsoc.org/doi/pdf/10.1175/JCLI-D-16-0666.1); does that apply do these models as well?

We do actually share the reviewers concern. However, please keep in mind that CloudSat remains the only source of snowfall estimates covering nearly the entire Arctic. Although we do not have multidecadal data, we believe about 10 years of data are sufficient enough to capture the first order features, seasonality and spatial variability in snowfall over the Arctic. This has therefore been the focus of our evaluations. The climate models in question are being used for the next IPCC assessments and it therefore becomes necessary to work with the data we currently have to understand the model performance in simulating snowfall.

With regard to the concern if the individual snowfall events can impact monthly snowfall estimates, we show below 1) the spatial distribution of the total number of snowfall pixels available at the original 1x1 deg grid, accumulated for each season studied here from 2006 to 2014,  and 2) monthly time series of the number of snowing pixels accumulated over the three

selected regions shown in Fig. 5.
It can be safely concluded that the monthly averages during the SON, DJF och MAM months are not represented by just a few strong events.

[Figure]

Two two figures are now added as Supplementary information in the revised version of the manuscript.

In Figure 7 of the manuscript, we have already investigated the distribution of snowfall in models versus CloudSat. We have expanded the discussion in the revised version, especially in conjunction with the recent study that the reviewer mentioned.

3. Choice of models: why were only these models chosen? Why only ocean-forced models and not fully-coupled models? Why is opted to not use a COSP-style radar simulator in the models? How does that impact the results? More details on the PRIMAVERA project are necessary. In any case, if only AMIP-style models are chosen, the authors should use the overlapping period between observations and models only.

This study was carried out as part of the PRIMAVERA project and the main aim was to evaluate those GCMs that would participate in the next IPCC assessments. The following line about the

PRIMAVERA project  is added in the revised manuscript. "Hence, the main aim of this study is to evaluate the HighResMIP (High Resolution Model Intercomparison Project) simulations for CMIP6 (Haarsma et al., 2016) under the PRIMAVERA (PRocess-based climate sIMulation: AdVances in high resolution modelling and European climate Risk Assessment) project. This project is a  European Union H2020 project wherein a total of 7 state of the art models are run at varying resolutions to understand the impact of resolution on different global climate processes.".

As far as we know, unfortunately there isn't any COSP-like simulator currently available that can be applied for comparing snowfall.

4. Methods, results, and discussion are mixed throughout the paper, which is somewhat confusing for the reader. A conclusion section with a long list reads like notes from a presentation, and should be converted a flowing text instead. More discussion should be added; why are these results important? How do these compare to other literature? What can we learn from it? What processes are lacking and/or inadequate in models?

The conclusion section is revised to provide more clarity on these issues.

Minor comments (P=page, L=line)
P1, L2: surface radiation budget
This is edited in the manuscript.

P1, L20: How would incoming radiation cool the surface?
'... thereby cooling the surface' has been removed.

P2, L34: irrelevant for this paper; Alpine snowfall is much more related to mid-latitude atmospheric dynamics, as well as orographic snowfall.
Indeed this study looks into the Alpine snowfall. It is however included as it carried out a model intercomparison between snowfall observations and a wide scale of models.

P3, L10: What is this project and what is its aim, and how does this paper fit in that?
The following sentences are added to the manuscript: Hence, the main aim of this study is to evaluate the HighResMIP CMIP6 simulations under the PRIMAVERA (PRocess-based climate sIMulation: AdVances in high resolution modelling and European climate Risk Assessment) project. This project is a European Union H2020 project wherein a total of 7 state of the art models are run at different resolutions to understand the impact of resolution on different global climate processes.

P3, L15: what is a snowfall flux? Mass or energy?
The snowfall flux is in kg/m2/s. The models output the snowfall estimates as snowfall flux, whereas, the CloudSat retrievals output snowfall accumulation. Hence, the model output are converted to snowfall accumulation for fair comparison. This is updated in the manuscript.

P3, L21: later, you discuss results from 2006-2015. Try to be clear
This is clearly addressed in the text. Though these simulations run from 1980-2015, for the computation of percentiles, a ten year period overlapping the observational period is considered. However, for the AO variability calculations, the whole period is considered. This is now clarified in the text.

P3, L27: possibly?
The word 'possibly' refers to the fact that some of the HighResMIP models used in this study consider graupel.

P4, L19: how do you end up with ten seasons? Clarify.
Sorry, that was a typo. I meant 'the' instead of 'ten'.

P4, L20 to P5, L8: these are methods, not results. Consider changing your section titles to improve structure (Section 2: Data and Methods; Section 3: Results; Section 4: Discussion and Conclusions)
The revised manuscript is organized accordingly.

P5, L16: larger difference
P5, L29: extent
The above changes are done in the revised manuscript.

P5, L31: what does 'negatively skewed' mean?
When the models tend to overestimate the observed response, so, in the Gaussian curve, the model response would be towards the right of the observed line (as can be seen in Fig.7 for p10; the black line that corresponds to the CloudSat data lies on the x-axis), then the distribution is said to be negatively skewed.

P6, L9 and beyond: This belongs to the discussion section.
This is moved to the discussion section.

Figures: too small; add units and indicate seasons in Figure itself. Add sublabels (e.g. a, b, etc) to refer to in text.
The units are now given in the figure itself. The seasons are already mentioned in the figure. The sublabels, (a)-(d) are given for the models in the revised manuscript.

P10, L32: this is the main issue of this paper; it should be added to the discussion section and reflected/expanded upon – see Main Issue 2
The discussion section is revised in the manuscript.

P12, L5: this is part of Methods, and does not fit here
This is moved to the 'Methodology' section in the revised manuscript.

P15, L1-13: Methods.

Section 6: This has not been clearly introduced in the intro and feels obsolete. Why would this be relevant to this paper? Why not also look at impact of NAO, Arctic sea ice extent, GBI, and other indices?

The importance of AO and its relevance is mentioned in the Introduction section in the revised version, instead of mentioning it directly in Section 6. As was mentioned in Section 6, AO is the dominant mode of natural variability in the Arctic and has large impact on precipitation variability (mainly in the form of snow during the polar winters). Therefore, it is important that the models capture this response of snowfall to AO, at least to a first degree, to be able to reasonably represent Arctic climate variability.

We preferred to investigate AO over NAO, mainly because while NAO is regional (mainly affecting the Atlantic sector), the AO is considered to have an Arctic wide impact.

P17, L4: I would argue that this is definitely not 'a wide range of models'

We look into 4 different models run at two resolutions each. Hence, the term 'wide range' of models.